# The Meta-Intelligent Child: Validating the MKIT as a Tool to Develop Metacognitive Knowledge in Early Childhood

**DOI:** 10.3390/jintelligence13110149

**Published:** 2025-11-17

**Authors:** Onciu Oana, Prisacaru Flavia

**Affiliations:** 1Faculty of Psychology and Educational Sciences, Department of Educational Sciences, “Alexandru Ioan Cuza” University of Iași, 700554 Iași, Romania; 2Faculty of Psychology and Educational Sciences, Department of Psychology, “Alexandru Ioan Cuza” University of Iași, 700554 Iași, Romania; flavia.prisacaru@student.uaic.ro

**Keywords:** metacognitive knowledge, metacognitive tool, stepped-wedge, early childhood, empirical validation

## Abstract

This article presents and validates the Metacognitive Knowledge Intervention for Thinking (MKIT)—an educational framework designed to assess and develop domain-general metacognitive knowledge (MK) in children aged 5 to 9. Moving beyond traditional approaches that examine metacognition within isolated subject areas, this research reconceptualizes MK as a transferable learning resource across content domains and developmental stages. Moreover, by employing a stepped-wedge design—a rigorous but rarely used approach in education—the study introduces a methodological advancement. Simultaneously, MK is operationalized through an ecologically valid and developmentally appropriate format, using visually engaging stories, illustrated scenarios, and interactive tasks integrated within classroom routines. These adaptations enabled young learners to engage meaningfully with abstract metacognitive concepts. Therefore, across three interconnected studies (N = 458), the MKIT provided strong psychometric evidence supporting valid inferences about metacognitive knowledge, age-invariant effects, and substantial gains among children with initially low MK levels. In addition, qualitative data indicated MK transfer across contexts. Thus, these findings position MKIT as a scalable tool, supported by multiple strands of validity evidence, that makes metacognitive knowledge teachable across domains—offering a practical approach to strengthening learning, reducing early achievement gaps, and supporting the development of core components of intelligence.

## 1. Introduction

Metacognition is widely recognized as a core dimension of self-regulated learning, integrating knowledge, monitoring, and control processes applied to one’s own cognitive activity. Although it is often introduced through the heuristic phrase “thinking about thinking” ([16]), such reduction obstructs the theoretical complexity, as well as the operational span of the construct.

In relation to intelligence, metacognition serves as a moderator by shaping how individuals apply their cognitive resources in tasks requiring learning and problem-solving. While intelligence provides the foundational capacity for reasoning, memory, and information processing, metacognition governs the effective use of these abilities through processes such as self-awareness, monitoring, control, and self-evaluation ([29]). Building on this, the framework of cognitive science positions metacognition as a second-order cognitive system. That is, a system that operates upon primary cognitive activities such as memory, comprehension, reasoning, and problem-solving. Specifically, metacognition enables individuals to access internal representations of their mental states, evaluate the adequacy of their thinking, and make strategic decisions to adjust and optimize performance ([15]). This mechanism is neither peripheral nor optional in human cognition; rather, it is accepted by modern research as a core process in adaptive cognitive development performance, expertise ([42]), and as an indisputable component of all lifelong learning ([22]; [19]; [37]).

Alternatively, from the perspective of the developmental frameworks, empirical evidence theorizes metacognition to be correlated with both cognition and theory of mind (ToM). This invariable interdependence between metacognition, cognition, and theory of mind is both conceptual as well as developmental. The convergence reflects a shared conceptual space—not merely as taxonomic distinctions, but as epistemological and terminological differentiations among constructs. More precisely, the interdependency is evident in the emergence of cognitive self-regulation, reflection, and social understanding. This is due to the fact that these cognitive phenomena systematically integrate a sequence of interrelated components and mechanisms that, taken together, build the emergence of higher-order cognitive functions ([17]). Thus, primally the cognition comprises first-order operations—such as perception, encoding and recall of information, inferential reasoning, language processing, and learning mechanisms—that underlie human interaction with the environment ([18]). Secondly, metacognition, by contrast, operates at a higher level—it involves the individual’s ability to monitor, evaluate, and regulate the cognitive processes through reflective and strategic operations ([36]). Finally, the theory of mind (ToM) enables the interpretation and anticipation of others’ cognitive states in social and learning contexts ([44]).

Furthermore, when examined within the framework of functional roles and epistemological value, metacognition reveals a recursive quality within its structure that accentuates a fundamental tension: in order to be effective, a cognitive system must be capable of both monitoring and regulating itself. This requires the capacity to evaluate the current state of the system and the extent to which progress is being made toward achieving its objectives. Equally important, the system must be capable of adapting its behavior based on these evaluations in order to pursue goals in the most efficient manner possible ([27]). In this context, metacognition functions as the foundational system that both reflects and shapes the maturation of higher-order thinking across developmental stages.

Yet such functioning is only possible insofar as it is grounded in *metacognitive knowledge*—a prerequisite that enables the coordination of monitoring, evaluation, and cognitive control mechanisms. As both a conceptual and functional anchor, it constitutes the basis upon which higher-order thinking, strategic awareness, and adaptive control are progressively constructed and developed. In other words, metacognitive knowledge refers to the awareness and understanding of one’s own cognitive architecture, including knowledge about strategies, tasks, and personal capabilities ([39]). Without this basic layer, subjects and individuals are deprived of the frame of reference necessary to interpret, evaluate, and regulate one’s own cognitive activity in order to make the cognitive system efficient.

Evidently, this form of knowledge is not limited exclusively to declaratively expressed cognition, but involves a multifactorial understanding that is sensitive to context and includes (a) knowledge about the person: awareness of one’s own and others’ cognitions, cognitive limitations, or capacities; (b) knowledge about the task: recognition of the conditions and complexities imposed by a task; and (c) knowledge about strategies: understanding the adequacy, availability, and effectiveness of different cognitive, learning, or problem-solving approaches. Finally, it is essential to note that metacognitive knowledge is neither abstract nor static—it is concretely activated, context-influenced, and progressively developed and refined through interaction and experience. Although in early childhood it may manifest itself in a situation-dependent and fragmented manner (for example, through spontaneous self-corrections, selective attention, or statements such as “that’s hard”), as children are exposed to reflective discourse, feedback, and meaningful learning experiences, metacognitive knowledge consolidates over time into a generalizable and strategic understanding. From this perspective, metacognitive knowledge—especially in its general, transferable form—emerges not as a by-product of cognitive growth but as a core component that actively shapes it. Within the developmental framework, the ages of 5 and 9 constitute critical developmental benchmarks for metacognitive growth. Around age 5—marking the late preoperational stage—metacognitive knowledge begins to emerge in explicit yet fragile forms: children may recognize difficulty or uncertainty, yet their awareness of self, task evaluation, and strategy usage remain superficial and heavily scaffolded ([7]). Without timely intervention at this stage, these fragile cognitive representations risk solidifying into entrenched patterns that constrain subsequent learning. By age 9—during the concrete operational stage—metacognitive knowledge typically becomes more consolidated and differentiated, enabling deliberate regulation and strategic processing ([34]). These developmental anchors are therefore critical not only for refining theoretical models but also for guiding interventions aimed at fostering robust metacognitive skills or addressing lingering gaps before children encounter the more demanding cognitive and academic milestones that lie ahead. This conceptual structure of metacognitive knowledge generates increasing theoretical and methodological demands to further clarify the mechanisms underlying its development, as well as the ways in which it can be reliably identified, systematically developed, and validly measured during its emergent stages—particularly in early childhood.

The most recent methodological challenges are largely defined by the difficulty of capturing and assessing this developmental trajectory with both precision and developmental sensitivity. This is important especially because, in early childhood, the person-related subcomponent is predominantly characterized by global, often idealized, representations of one’s own cognitive efficiency. Although incipient, this form of metacognitive knowledge can be captured through semi-structured interviews or self-prediction protocols or observations focused on spontaneous meta-comments ([26]). Regarding task knowledge, the data show an early emergence of the capacity for contextual differentiation—children begin to discriminate levels of difficulty and anticipate implicit demands. Narrative scenarios and vignette-based tests are well suited to access these incipient interpretive structures ([33]). The latter, used to crystallize knowledge and essential for cognitive autonomy, remains a strategy-related subcomponent. In this direction, visually assisted “think-aloud” sequences, strategic choice tests in semi-directed games, and reflective behavior coding grids provide valid access to the active strategic repertoire and associated metacognitive justifications ([41]). To complete the metacognitive profile, adapted versions of standardized instruments (MAI, Jr. MAI) can be used, provided that prior lexical and symbolic adaptation is made ([35]). Equally, a promising frontier in this field lies in longitudinal and mixed-methods designs that capture trajectories of metacognitive knowledge over time, particularly in naturalistic learning environments such as classrooms and homes ([5]). These models can reveal how metacognitive knowledge develops as a function of individual differences (e.g., executive function, temperament), contextual influences (e.g., quality of instruction, parental discourse), experiential factors (e.g., error experiences, quality of feedback), and guided developmental interventions (e.g., structured training or scaffolding) that actively promote metacognitive understanding. Finally, beyond structural empirical efforts to trace the progression of early development, another research concern emerges. Namely, modern contemporary empirical studies increasingly align the investigation of metacognitive knowledge strictly within domain-specific contexts. This trend is evident in research focusing on areas such as reading, mathematics, and science ([46]), where metacognitive assessment is typically embedded in tasks that are closely tied to the specific cognitive demands of each domain. As a result, the instruments used and the data collected tend to be highly contextualized and task-dependent—for instance, comprehension monitoring tasks in reading ([6]), or strategic behavior observed during structured mathematical problem-solving ([8]). While this approach has advanced our understanding of metacognition in authentic learning contexts, it also raises concerns about the transferability and generalizability of the findings across domains. Taken together, although the domain-specific approach remains the most relevant method within curricular educational perspective, it points to new lines of research—particularly in designing interventions that address general and transversal dimensions of metacognitive knowledge ([43]).

For the present study, we selected ages 5 and 9 as methodological anchors. This choice is grounded, first, in the theoretical canon of developmental psychology and second, the choice carries direct practical implications, as interventions at age 5 may serve to prevent later metacognitive difficulties, while interventions at age 9 can strengthen the necessary basis for the further development of higher-order metacognitive capacities ([25]; [11]).

## 2. The Present Research

The present research introduces the Metacognitive Knowledge Intervention for Thinking (MKIT), a framework designed to assess and enhance metacognitive knowledge (MK) in children aged 5 and 9. Despite growing interest in early metacognition, the most recent studies continue to be shaped by several foundational limitations. First, much of the assessment work in young children relies on instruments originally designed for older populations, with minimal developmental adaptation. Simplified versions of tools like the Metacognitive Awareness Inventory (Jr. MAI), when used with children aged 5 to 9, often still place excessive verbal and cognitive demands on respondents ([45]). Second, the field remains methodologically constrained. A large number of studies continues to rely on cross-sectional or basic pre–post designs that lack rigorous intermediate checkpoints—thereby weakening causal claims regarding intervention efficacy and persistence ([12]; [42]). Moreover, while a few studies have explored observational and qualitative methods with some success, the use of ethical, scalable, and analytically robust experimental designs in ecologically valid settings is exceedingly rare ([14]). This gap is particularly problematic in early childhood research, where the feasibility of real-world implementation and equitable access to intervention are crucial. As a result, to our knowledge, MKIT is the first framework of this kind, with empirical evidence supporting valid interpretations of its outcomes.


*The Validation Framework*


Following [32]’s ([32]) unified framework and [24]’s ([24]) argument-based approach, we did not treat validity as a collection of separate “types”, but rather as a single construct supported by multiple strands of evidence. From this perspective, the present research conceptualizes validity as the degree to which evidence and theory support the interpretations of test scores for their intended uses. Consequently, we frame our analyses in terms of (a) content and response process evidence (translation, cultural adaptation, cognitive interviews); (b) Evidence about internal structure (factor analyses, reliability indices); (c) relations to other variables (correlations with MAI, developmental differences between age groups); (d) generalization evidence (replication across studies and designs); and (e) consequences of testing (instructional utility and observed learning gains).

Evidence was gathered across the three studies as follows. Content and response process evidence was ensured through rigorous translation, cultural adaptation, and expert evaluation of item relevance, complemented by child cognitive interviews confirming comprehensibility. Structural evidence was provided by exploratory and confirmatory factor analyses supporting a unidimensional solution, together with internal consistency indices (Cronbach’s α, McDonald’s ω). Generalization evidence emerged from consistent findings across the pilot and quasi-experimental studies. Relations-to-other-variables evidence was demonstrated by a strong positive correlation with the Romanian adaptation of the MAI (r = 0.738, *p* < .01) and systematic age group differences (5 vs. 9 years). Finally, consequential evidence came from instructional studies, where systematic improvements in metacognitive knowledge were observed following MKIT activities, highlighting the usefulness of McKI scores for guiding classroom practice.

The program of research comprised three interrelated studies, each aligned with a strand of the validity argument: Study 1: Adaptation and collection of psychometric evidence supporting McKI inferences (focus on internal structure, response processes, relations to MAI). Study 2: Pilot implementation of MKIT using a stepped-wedge design (focus on generalization and intervention effects across age groups and baseline levels). Study 3: Large-scale quasi-experimental trial integrating quantitative and qualitative evidence (focus on generalization, consequences of testing, and transfer of MK across contexts). Taken together, these studies provide complementary strands of evidence that, when integrated, support the validity argument for interpreting and using MKIT scores in early childhood classrooms.


*The Instrumental Contribution*


In Study 1, we developed MKIT through both structural and semiotic adaptations, resulting in a tool that is developmentally appropriate and pedagogically integrated, thus suited for research as well as classroom use in early childhood. MKIT unifies two previously validated instruments into a coherent framework that combines assessment and instructional components, each visually adapted to reduce linguistic load and increase accessibility (see Appendix A). The assessment component is based on the Metacognitive Knowledge Interview for Children (McKI), a semi-structured interview of 11 scenario-based items designed to elicit metacognitive responses in young learners ([28]). The instructional component, which also served as a basis for qualitative analyses, draws on 16 items from the Metacognitive Awareness Inventory (MAI), adapted and validated for Romanian early childhood populations ([23]).


*The Methodological Contribution*


Study 2 piloted the full MKIT intervention using a stepped-wedge experimental design, assessing its short-term impact on metacognitive knowledge and examining whether outcomes varied as a function of age or baseline MK levels. The adopted design is still rarely applied in early education even if it is acknowledged for its ethical robustness and internal validity ([31]). From an epistemological perspective, the design aligns with a pragmatic-constructivist paradigm, which values not only causal inference but also the generation of contextually situated and practically relevant knowledge. By capturing changes over time and across educational settings, the stepped-wedge design allowed researchers to address more complex research questions; not only whether the intervention worked, but also how, for whom, and under what conditions—a central concern in contemporary educational research and policy ([3]).


*The Ecological and Contextual Contribution*


Across all three studies, the intervention was implemented entirely within regular public-school settings, integrated seamlessly into existing curricular routines. This contextual anchoring addresses the long-standing critique that most metacognitive research is either detached from practice by taking place in online contexts or by being limited to clinic laboratory experiments. By situating MKIT in real classrooms within school schedules, the framework maximizes both feasibility and scalability—two critical conditions for real-world educational impact ([4]).

Together, these objectives provide a focused and empirically grounded contribution to the literature by offering evidence for MKIT as a reliable and valid tool for studying metacognitive knowledge in early childhood. In sum, these design choices signal a deliberate and research-grounded response to the most pressing limitations in early metacognitive research. To conclude, MKIT not only advances methodological rigor but also reframes what ethically grounded, developmentally appropriate, and pedagogically meaningful metacognitive intervention can look like in the early years—setting a precedent for future intervention and policy design in educational contexts.

## 3. Materials, Methods and Initial Findings

### 3.1. Study 1: The Adaptation and Initial Validation of the Assessment Tool

The initial phase of the study was dedicated to the translation, cultural adaptation, and validation of the McKI for the Romanian early childhood population, given that the MAI had already been previously adapted for the Romanian context. As such, this phase concentrated on aligning the McKI instrument with the linguistic and developmental characteristics of the target population. Permission for the translation and use of the McKI was granted by the original author.

#### 3.1.1. Participants and Research Setting

To support instrument validation, 100 children were recruited for the empirical phase of the study. The sample was evenly distributed across two age groups: 50 five-year-olds and 50 nine-year-olds. Gender distribution included 41 boys and 59 girls. All participants were enrolled in public kindergartens or primary schools in Romania and represented a range of socio-educational backgrounds. Inclusion criteria required that children were monolingual Romanian speakers and had no reported developmental delays, based on parent and teacher reports.

At this stage, only the McKI instrument was examined. A single pre-test administration was conducted to evaluate its psychometric properties, without implementing the intervention component. The data collection took place in familiar, group-based educational environments during regular class hours. Equally, all procedures were conducted individually, in quiet and distraction-free settings, by the researcher using standardized protocols. This approach minimized the risk of peer influence or response contamination and ensured consistency across participants.

Ethical approval was granted by all institutional review boards overseeing the participating schools, and written informed consent was obtained from parents or legal guardians of all participants. Age-appropriate assent procedures were also implemented, using visual and verbal explanations to ensure that children understood the voluntary nature of participation and their right to withdraw at any time.

#### 3.1.2. Design, Procedure, and Validity Expectations

During the adaptation process of the McKI, particular attention was paid to simplifying and clarifying item phrasing in order to align with the linguistic and cognitive capacities of young children ([21]). A panel of three bilingual developmental psychologists independently reviewed all items for conceptual clarity and age appropriateness ([9]). Afterwards, qualitative interviews were conducted with 60 Romanian children (30 aged 5; 30 aged 9) to examine item comprehensibility and linguistic clarity. This group of children was distinct from the main validation sample used later in the subsequent statistical analyses of Study 1. By doing so, we ensured that feedback on item phrasing did not influence subsequent psychometric analyses.

Moving forward, minor lexical adjustments and procedural scaffolds were incorporated based on this feedback. Several items were linguistically refined to improve specificity and situational relevance. For instance, the item “*Do you think something was difficult?*” was rephrased as “*Do you think that something was difficult in this task?*”, to guide children’s reflections toward a specific metacognitive knowledge dimension and avoid overly broad or ambiguous interpretations. More broadly, abstract formulations were replaced with concrete, metacognitive anchored alternatives. Finally, items related to help-seeking and strategy selection were supplemented with age-appropriate scaffolds, such as concrete examples or clarifying phrases.

Specifically, we anticipated the following: (*a*) Evidence about internal structure—Items would demonstrate sufficient inter-item correlations to justify factor analysis, yielding a unidimensional latent structure (*Hypothesis 1*). (*b*) Reliability evidence—Internal consistency indices (α, ω) would meet or exceed accepted thresholds of 0.70 (*Hypothesis 2*). (*c*) Relations to other variables—McKI scores would correlate positively with the Romanian MAI, reflecting conceptual overlap and supporting relations-to-other-variables evidence (*Hypothesis 3*). (*d*) Developmental differentiation evidence—McKI would distinguish between 5- and 9-year-olds, with ROC analysis confirming classification accuracy above chance (*Hypothesis 4*).

Analyses included exploratory factor analysis (EFA), reliability estimation (Cronbach’s α, McDonald’s ω, and item–total correlations), correlations with the MAI, and group comparisons supplemented by ROC analysis.

#### 3.1.3. Interim Findings


*Internal Structure Evidence*


In line with the prediction of Hypothesis 1, the data met criteria for factor analysis, KMO = 0.735 (adequate sampling), and Bartlett’s test was significant, *χ*^2:^ = 208.87, *p* < .001, confirming sufficient inter-item correlations for factor extraction. Furthermore, EFA using principal axis factoring supported a one-factor solution, explaining 23.82% of the variance, consistent with the hypothesized metacognitive knowledge construct. Most items loaded meaningfully on the primary factor, confirming the prediction that the majority would exhibit substantial loadings (≥0.40).

Notably, items McKI 11, 6, 9, 8, and 5 showed loadings above 0.50, indicating strong contributions to the latent construct. Items McKI 1, 2, 3, and 4 had low communalities (<0.20), but were retained for theoretical significance. In addition, no cross-loadings were observed, reinforcing the internal coherence of the one-factor solution.

Additionally, considering the preliminary stage of instrument development and constraints related to sample size (N = 100), only exploratory factor analysis was conducted at this point. Confirmatory factor analysis (CFA) was conducted in Study 3, where a larger and independent sample permitted a more robust examination of the instrument’s latent structure.


*Reliability Evidence*


Consistent with the expectations in Hypothesis 2, internal consistency was acceptable, *α* = 0.76, *ω* = 0.76, and average inter-item correlation = 0.22 (within the 0.15–0.50 range), indicating moderate item interrelatedness without redundancy. Given the recognized limitations of Cronbach’s alpha as a reliability index ([40]), we also report McDonald’s omega, in line with recent recommendations for more robust reliability estimation ([30]). Therefore, evidence of internal consistency was acceptable. Additionally, item-wise diagnostics supported the scale’s cohesion: most items contributed positively, and removing any item had minimal impact on *α*.


*Convergent Validity Evidence*


To assess convergent validity, correlations were computed between scores on the McKI and the adapted Metacognitive Awareness Inventory (MAI), which similarly targets metacognitive knowledge in young children. Thus, in examining Hypothesis 3, McKI scores correlated strongly with the MAI (*r* = 0.738, *p* < .01), supporting convergent evidence by demonstrating substantial conceptual overlap between the two instruments.


*Discriminant Validity Evidence*


Supporting Hypothesis 4, the scale distinguished between age groups as expected: older children (*M* = 1.25) scored significantly higher than younger peers (*M* = 0.91), *t*(98) = −9.04, *p* < .001, 95% CI [−0.41, −0.26], providing evidence of discriminant validity and scale’s sensitivity to age-related in metacognitive understanding. Ultimately, ROC analysis confirmed this distinction, yielding a classification accuracy of 96% (*p* < .001), demonstrating the instrument’s strong capacity to distinguish between developmental levels at a statistically robust level.

These findings establish the structural evidence of validity for the adapted McKI and justify its use in subsequent intervention phases, as examined in Study 2.

### 3.2. Study 2: The MKIT Pilot Trial

Study 2 piloted the integrated intervention system—comprising both assessment and instructional components (McKI and MAI, respectively). In this exploratory pilot trial, we implemented a stepped-wedge randomized design (SW-CRT) to rigorously assess the intervention effects, age-based differences in responsiveness, the persistence of outcomes over time, and differential gains based on participants’ baseline metacognitive knowledge.

Equally, we translated 19 MAI items ([23]) into illustrated stories and age-adapted metacognitive activities. By structurally rethinking item formulation and introducing semiotic scaffolds grounded in age specificity and everyday learning contexts, this study moves beyond linguistic simplification to achieve conceptual precision and developmental fidelity. In doing so, it contributes not only to the validation of a specific tool, but to the broader endeavor of designing reliable measurement systems that are epistemologically appropriate for early cognitive development. This semiotic approach enabled the transformation of the MAI from a verbally dense, adult-oriented inventory into a multimodal instrument, where meaning is co-constructed through text, image, and context. As such, the adapted items supported both comprehension and self-expression, allowing children to reflect on their thinking even in the absence of advanced verbal reasoning. Thus, the MKIT served as both a research instrument and a developmental window into how young children articulate and reflect upon their own thinking processes—a capacity central to effective self-regulated learning and long-term academic adaptation.

#### 3.2.1. Participants and Research Setting

To support the pilot evaluation of the integrated intervention system, 80 children were recruited from 12 educational clusters (C1–C12). The sample was evenly distributed across two age groups: 40 five-year-olds and 40 nine-year-olds. Gender distribution included 39 boys and 41 girls. Due to the stepped-wedge structure, each group served as its own control at different phases of the study, enhancing internal validity by combining both within- and between-cluster comparisons. All participants were enrolled in public kindergartens or primary schools within the same geographical region, representing diverse socio-educational backgrounds. Inclusion criteria mandated that children be monolingual Romanian speakers with no reported developmental delays, as verified by parent and teacher reports. Recruitment was conducted through institutional partnerships, and participants were randomly assigned to clusters following a stepped-wedge design. Data were collected at 7 time points (waves) across the intervention period, enabling longitudinal tracking of metacognitive knowledge development. The sample size, while moderate, was determined based on pilot study constraints and the stepped-wedge design, which maximizes statistical power by utilizing both within- and between-cluster comparisons.

#### 3.2.2. Design, Procedure, and Validity Expectations

The stepped-wedge intervention was delivered in a group format across 6 consecutive weeks, within the children’s regular classroom environments. Each week consisted of 4 metacognitive activity sessions conducted from Monday to Thursday, covering four recurring types of tasks: illustrated stories, semi-structured games, writing and drawing activities, and strategic reasoning exercises. These activities were designed to foster metacognitive knowledge. Fridays were reserved for data collection, during which individual interviews were conducted with each child. This weekly structure allowed for a clear separation between instructional and assessment components, minimizing potential interference between the two. To prevent contamination effects and social desirability bias, all assessments were conducted on an individual basis in distraction-free environments, ensuring consistent and independent responses from each participant. To ensure fidelity of implementation, the intervention was administered by the principal investigator using a standardized delivery and observation protocol applied uniformly across all clusters. Data collection occurred across 7 assessment waves. Each administration followed a standardized format and lasted approximately 17–20 min per child, enabling the consistent tracking of metacognitive development over time. Specifically, 12 educational clusters (C1–C12), divided by age (children aged 5 and 9), were included in a protocol with 7 successive measurement moments. Each week, two clusters were introduced to the intervention, according to a predefined rotation schedule. The corresponding procedure reflected this sequence, marking the transition points from the control to the treatment condition. While the pilot nature of the study entailed a moderate sample size (N = 80), the implementation followed high experimental standards, including the systematic application of individual measurements and temporal control. In addition, the intervention was carried out in real educational contexts, strengthening the ecological validity of the results.

Subsequently, the analytical approach adopted for this study was designed to examine the immediate and sustained effects of the MKIT intervention on children’s metacognitive knowledge, while accounting for contextual factors. Given the longitudinal design with repeated measurements across 7 waves and the nested structure of data (children within educational clusters), mixed-effects models were employed to appropriately model intra-individual change and inter-individual variability.

The hypotheses of this study addressed multiple sources of evidence supporting validity. (*a*) Consequential evidence—It was expected that the introduction of the MKIT intervention would yield significant immediate gains in children’s metacognitive knowledge, compared to the pre-intervention period. Demonstrating short-term improvements provides evidence that the use of the instrument can positively impact educational outcomes, reinforcing consequential evidence by highlighting the potential of the instrument to reduce developmental disparities (*Hypothesis 1*). (*b*) Generalization across developmental subgroups—The intervention effect was expected to differ by age, with a significant interaction anticipated between age group (5-year-olds vs. 9-year-olds) and treatment efficacy. Such age-based differences would provide generalization evidence by demonstrating the robustness and developmental sensitivity of score interpretations across distinct subpopulations (*Hypothesis 2*). (*c*) Generalization across time—Beyond immediate effects, we anticipated that improvements in metacognitive knowledge would persist or consolidate across subsequent measurement waves. A positive linear post-intervention trajectory was hypothesized, with possible non-linear trends reflecting curvilinear growth. Evidence of stability or strengthening effects over time would support the generalization aspect of validity across temporal contexts (*Hypothesis 3*). (*d*) Relations-to-other-variables and consequential evidence: Finally, it was expected that children with lower baseline metacognitive knowledge would benefit disproportionately from the intervention, showing greater relative gains compared to higher-performing peers. This compensatory pattern would provide relations-to-other-variables evidence (sensitivity to initial ability differences) while also reinforcing consequential evidence by highlighting the potential of the instrument to reduce developmental disparities (*Hypothesis 4*).

To statistically examine these validity-driven hypotheses, mixed linear models (MLMs) were employed.

#### 3.2.3. Interim Findings


*Evidence of Immediate Intervention Effects*


To test Hypothesis 1, descriptive statistics results indicated that scores increased in post-intervention from *M* = 10.19 (*SD* = 2.96) at pre-test to *M* = 20.68 (*SD* = 1.89) at post-test, with a mean difference of −10.49 points (*SD* = 3.16, *SE* = 0.13). The effect size was large, *d* = 1.94, indicating a substantial educational impact.

The findings revealed a strong and highly significant intervention effect, *t*(559) = −78.51, *p* < .001, suggesting that treatment condition accounted for a substantial proportion of the variance in post-intervention outcomes. In line with Hypothesis 1, children demonstrated higher levels of metacognitive knowledge following participation in the intervention.


*Evidence of Age Effects*


Hypothesis 2 predicted greater intervention gains among older children (age 9) compared to younger children (age 5). This hypothesis was not supported. The comparison revealed no statistically significant differences in MK scores by age group, *F*(1, 556) = 0.011, *p* = .917. Thus, the intervention effect was consistent across age levels, suggesting developmental robustness in MKIT responsiveness.


*Evidence of Age × Intervention interaction*


To examine Hypothesis 3, which proposed an interaction between age and intervention efficacy, mixed-effects modeling assessed whether improvement in MK scores varied by age. The Age × Intervention interaction was not statistically significant, *F*(1, 556) = 1.620, *p* = .204, indicating that children across age groups improved similarly.


*Evidence of Group × Time Interaction*


Hypothesis 4 posited that intervention gains would be maintained or even strengthened over subsequent measurement waves. Longitudinal analysis across seven time points showed that children in the intervention group experienced a steady and cumulative increase in MK scores, from *M* = 0.93 (*SD* = 0.27) at Wave 1 to *M* = 1.78 (*SD* = 0.19) at Wave 7. In contrast, participants in the control group exhibited only minimal gains over the same period, with clearly divergent performance trajectories between groups.

To further examine the persistence of effects, a linear regression analysis was conducted using baseline MKI scores as a predictor of final outcomes. The results confirmed a significant predictive relationship, *F*(1, 558) = 26.614, *p* < .001, with an *R*^2^ of 0.046 (adjusted *R*^2^ = 0.044), and a standard error of the estimate of 1.856.

The regression model confirms that initial metacognitive performance was a significant, though modest, predictor of final scores. Specifically, the MKI initial score had a positive effect, *β* = 0.137 (SE = 0.026), standardized *β* = 0.213, *t* = 5.159, *p* < .001, while the intercept was *β* = 19.284 (*SE* = 0.281), *t* = 68.676, *p* < .001.


*Evidence of Baseline-Dependent Effects*


Hypothesis 5 predicted that children with lower baseline metacognitive scores would experience greater relative benefits from the intervention. This pattern was confirmed by a one-way ANOVA conducted using four baseline performance groups. The results revealed a statistically significant effect of baseline metacognitive knowledge on gain scores, *F*(3, 556) = 250.97, *p* < .001, indicating that the level of improvement varied systematically by initial performance level.

### 3.3. Study 3: The MKIT Formative Experiment

Study 3 functioned as the confirmatory stage of the research program, aiming to provide generalization evidence and to test the large-scale effectiveness of the MKIT framework across a more diverse and extensive sample of children.

Specifically, the study pursued three complementary objectives, each aligned with a strand of Messick’s unified framework. First, it aimed to provide structural evidence by validating the psychometric structure of the adapted Metacognitive Knowledge Interview (McKI) through confirmatory factor analysis (CFA), thereby extending the exploratory findings from Study 1. Second, it sought to generate consequential and generalization evidence by evaluating the effectiveness of the MKIT intervention in producing developmental gains in metacognitive knowledge, using a quasi-experimental pre–post design with a control group. Third, it aimed to contribute relations-to-other-variables and transfer evidence through qualitative analyses examining whether children applied insights from MKIT activities in their reflective thinking, thus demonstrating generalized metacognitive awareness across domains.

#### 3.3.1. Participants and Research Setting

To support the large-scale impact evaluation of the entire study, a total of 278 participants were recurred. The randomized sample was divided into two groups: 218 children in the experimental group and 60 in the control group. Participants were drawn from two educational institutions in urban Romania—one kindergarten and one primary school—encompassing a diversity of socio-educational backgrounds. Each class was treated as a cluster of 30 students and retained in their natural educational setting to ensure ecological validity. Age was distributed across two strata: 135 children were 5 years old, and 143 were 9 years old. Gender was evenly represented across conditions with 156 girls and 122 boys. All children were monolingual Romanian speakers, with no reported cognitive or developmental impairments. The intervention was delivered in a group format during school hours over a 10-week period, while data collection—both quantitative and qualitative—was conducted individually, in quiet and distraction-free environments. The same standardized protocols and age-appropriate ethical procedures used in Studies 1 and 2 were employed to ensure consistency and data integrity.

#### 3.3.2. Design, Procedure, and Validity Expectations

Study 3 applied a quasi-experimental pre–post design with a control group, aiming to evaluate the structural validity of the adapted McKI, as well as the broader effectiveness of the MKIT intervention in authentic educational settings. The intervention was implemented over a 10-week period in the experimental group, using structured, group-based instructional sessions grounded in the MKIT framework. The control group received standard classroom instruction without any metacognitive training components. To measure change, all participants completed the McKI individually, both before and after the intervention, following standardized administration procedures. Additionally, a qualitative case study component was embedded to explore the internalization and developmental variability of metacognitive discourse. This component included in-depth interviews with three contrasting participants: one with the highest and one with the lowest post-test MK scores from the experimental group, and a third participant with an average score from the control group—each selected to illustrate distinct developmental trajectories.

Subsequently, the validation process was grounded in theoretically and empirically informed hypotheses. (*a*) Structural evidence—The adapted McKI was expected to exhibit a unidimensional factorial structure, reflecting a global construct of metacognitive knowledge. Confirmatory factor analysis (CFA) was anticipated to yield acceptable fit indices (*Hypothesis 1*). (*b*) Consequential and generalization evidence—Participants in the experimental group were expected to show significantly greater pre–post gains than those in the control group, indicating that MKIT participation enhances metacognitive knowledge in ecologically valid contexts (*Hypothesis 2*). (*c*) Generalization across developmental subgroups—The intervention’s impact was hypothesized to vary by age, with 9-year-olds exhibiting greater gains than 5-year-olds, consistent with developmental differences in metacognitive responsiveness (*Hypothesis 3*). (*d*) Relations-to-other-variables and transfer evidence—Children exposed to MKIT were expected to display richer, more integrated, and transferable metacognitive discourse during qualitative interviews, referencing reflective processes across person, task, and strategy dimensions. In contrast, control group children were expected to show more fragmented discourse, with limited evidence of strategic transfer or reflective reasoning (*Hypothesis 4*).

To statistically analyze the findings of these hypotheses, a multi-method analytical strategy was employed. Quantitative analyses assessed both the internal structure of the adapted metacognitive assessment instrument and the intervention’s impact on children’s metacognitive knowledge across time and groups. Statistical modeling accounted for developmental and contextual variables to examine whether the observed effects were consistent across age cohorts and experimental conditions. Complementing the quantitative data, a qualitative case study approach was used to explore the depth, coherence, and transfer of metacognitive discourse. This triangulation of methods provided a comprehensive framework for assessing both measurable gains and the internalization of metacognitive processes, offering robust insight into the developmental impact of the MKIT framework intervention.

All procedures and reporting followed the APA Journal Article Reporting Standards ([1]), which build on the updates to the JARS for quantitative research introduced in the 2018 revision ([2]), ensuring transparency and replicability.

#### 3.3.3. Interim Findings


*Evidence of Structural Validity*


To evaluate Hypothesis 1, which posited that the adapted McKI would exhibit a unidimensional latent structure consistent with theoretical expectations, confirmatory factor analysis (CFA) was conducted on pre-intervention scores. The results indicated an excellent model fit: *χ*^2:^ = 39.19, *p* = .677; CFI = 1.000; TLI = 1.005; RMSEA = 0.000 (90% CI [0.000, 0.033]); SRMR = 0.024. All items loaded significantly onto the single latent factor (λ = 0.84 to 1.18), with the exception of one marginal item (McKI11, *p* = .092), retained for theoretical relevance. The choice of fit indices and cutoff thresholds follows widely cited methodological standards and is consistent with more recent discussions on factor retention and model fit evaluation ([20]). Accordingly, the results confirm robust factorial evidence supporting valid score interpretations of the adapted McKI, thereby supporting Hypothesis 1.


*Evidence of Intervention Effects*


Hypothesis 2 proposed that children in the experimental group would demonstrate greater gains in metacognitive knowledge compared to the control group. A two-way mixed ANOVA revealed a significant main effect of time, *F*(1, 276) = 202.32, *p* < .001, *η*^2^ = 0.423, and a significant time × group interaction, *F*(1, 276) = 202.32, *p* < .001, *η*^2^ = 0.423. Post-test scores increased from *M* = 11.25 (*SD* = 6.26) to *M* = 14.54 (*SD* = 5.40) in the experimental group, while the control group remained unchanged (*M* = 7.47, *SD* = 1.83). Pairwise comparisons confirmed that the post-test difference between groups was statistically significant (Δ = 5.43, 95% CI [3.93, 6.92], *p* < .001).


*Evidence of Age Effects*


Hypothesis 3 posited that older children (9 years) would benefit more from the intervention than younger children (5 years). A separate mixed ANOVA with age as a between-subjects factor and time as a within-subjects factor revealed a significant main effect of time, *F*(1, 276) = 423.91, *p* < .001, *η*^2^ = 0.606. However, the presumption was infirmed since the age × time interaction was non-significant, *F*(1, 276) = 0.006, *p* = .940, *η*^2^ = 0.000. Both age groups showed similar improvements (5-year-olds: *M* = 9.93 to 12.52; 9-year-olds: *M* = 10.91 to 13.48), indicating equivalent developmental responsiveness.


*Evidence of Qualitative Transfer*


To test Hypothesis 4, which predicted that children in the experimental group would demonstrate greater depth, integration, and transfer in metacognitive discourse, a qualitative case analysis was conducted. Three cases were analyzed: two from the experimental group (one high-scoring, one low-scoring), and one control group with child matched on age and baseline McKI scores.

The results revealed that the high-scoring experimental participant exhibited rich, reflective, and strategy-based metacognitive talk, with explicit references to MKIT activities. The low scorer showed more fragmented and context-bound discourse, while the control child demonstrated minimal reflective content and no evidence of intervention-related transfer. These patterns support Hypothesis 4, illustrating that for these specific cases, MKIT facilitated the internalization and transfer of metacognitive knowledge into autonomous self-reflection.

## 4. Integrated Findings of Validity

The three interrelated studies collectively provide robust empirical evidence for the Metacognitive Knowledge Intervention for Thinking (MKIT) framework, affirming its psychometric integrity, developmental sensitivity, instructional efficacy, and ecological evidence of validity. The choice of fit indices and cutoff thresholds follows widely cited methodological standards and is consistent with more recent discussions on factor retention and model fit evaluation ([20]). Through a triangulated methodological design combining exploratory and confirmatory factor analyses, stepped-wedge randomized trials, and large-scale quasi-experimental evaluation, the MKIT system was shown to elicit measurable and sustained improvements in metacognitive knowledge (MK) among children aged 5 to 9.

### 4.1. Evidence for Structural Validity and Psychometric Integrity

The adaptation and validation of the Metacognitive Knowledge Interview for Children (McKI) provided a foundational psychometric basis for the MKIT framework. Exploratory factor analysis (EFA) conducted in Study 1 confirmed a unidimensional structure aligned with theoretical conceptions of MK as a cohesive construct. Sampling adequacy (*KMO* = 0.735) and a significant Bartlett’s test (*χ*^2:^ = 208.87, *p* < .001) supported the suitability of the data for factor analysis and illustrated in Table 1. Additionally, Table 2 illustrates the factor loadings. Specifically, most items—particularly McKI 5, 6, 8, 9, and 11—exhibited substantial factor loadings (>0.50), with no cross-loadings, confirming the instrument’s structural coherence.

Further, internal consistency was acceptable, with Cronbach’s alpha and McDonald’s omega both reaching 0.76, and average inter-item correlations (0.22) suggesting good reliability without redundancy.

These values are presented in Table 3, alongside item–total correlations and diagnostics.

Convergent validity was supported by a strong correlation between McKI and the adapted MAI instrument (*r* = 0.738, *p* < .01), as detailed in Table 4.

Discriminant validity was confirmed through significant age-based differences in MK scores (*M*_9_ = 1.25 vs. *M*_5_ = 0.91), *t*(98) = −9.04, *p* < .001, and through a classification accuracy of 96% derived from ROC analysis. These findings, illustrated in Table 5 and Table 6, demonstrate the instrument’s developmental sensitivity.

Finally, confirmatory factor analysis (CFA) conducted in Study 3 further supported these findings, yielding excellent model fit indices (*χ*^2:^ = 39.19, *p* = .677; CFI = 1.000; RMSEA = 0.000; SRMR = 0.024). The CFA loadings and fit statistics are summarized in Table 7. Together, these psychometric outcomes confirm that the McKI is structurally sound, developmentally valid, and psychometrically reliable for early metacognitive assessment.

### 4.2. Instructional Outcomes as Consequential Validity Evidence

The MKIT intervention led to consistently strong and statistically significant improvements in MK across multiple contexts. In the pilot trial (Study 2), children demonstrated significant MK gains immediately following intervention exposure. As shown in Table 8, descriptive statistics indicated that scores increased in post-intervention from *M* = 10.19 (*SD* = 2.96) at pre-test to *M* = 20.68 (*SD* = 1.89) at post-test, with a mean difference of −10.49 points (*SD* = 3.16, *SE* = 0.13).

The effect size was large, *d* = 1.94, indicating a substantial educational impact. The findings are detailed in Table 9, and revealed a strong and highly significant intervention effect, *t*(559) = −78.51, *p* < .001, suggesting that treatment condition accounted for a substantial proportion of the variance in post-intervention outcomes. Aligned with Hypothesis 1, children demonstrated higher levels of metacognitive knowledge following participation in the intervention.

Contrary to initial expectations of Study 2, age did not moderate the intervention effect; both five- and nine-year-olds showed comparable improvements. Thus, this assumption was not supported by the data. As Table 10 findings show, no significant differences emerged between the two age groups, *F*(1, 556) = 0.011, *p* = .917, indicating that the intervention was equally effective across developmental stages, highlighting its robustness across age levels.

Regarding the baseline metacognitive scores, children with lower initial levels benefitted more from the intervention. As shown in Table 11, gain scores decreased systematically across the four baseline performance groups, with the largest improvements observed among children in the low baseline group (*M* = −14.50, *SD* = 3.07) and the smallest among those in the very high baseline group (*M* = −6.55, *SD* = 1.79). A one-way ANOVA confirmed that this pattern was statistically significant, *F*(3, 556) = 250.97, *p* < .001. Bonferroni-corrected post hoc comparisons are illustrated in Table 12, and further indicate that all pairwise differences were significant at *p* < .001, confirming that initial performance level systematically moderated intervention gains.

These findings demonstrate a compensatory effect of MKIT: children with initially lower metacognitive knowledge benefitted disproportionately from the intervention. Such a pattern is educationally relevant, as it suggests that MKIT not only enhances metacognitive knowledge overall but also reduces initial disparities, thereby promoting equity in early learning contexts.

Furthermore, to evaluate the overall effectiveness of the MKIT intervention, Study 3 employed a larger, quasi-experimental design (N = 278). A mixed ANOVA revealed a significant time × group interaction, *F*(1, 276) = 202.32, *p* < .001, *η*^2^ = 0.423, confirming the instructional impact of MKIT. As shown in Table 13, the intervention group improved from *M* = 11.25 (*SD* = 6.26) at pre-test to M = 14.54 (*SD* = 5.40) at post-test, where the control group scores remained unchanged (*M* = 7.47, *SD* = 1.84). At baseline, pre-test scores differed between groups (Hedges’ g = 0.67, 95% CI [0.38, 0.96]). To account for this imbalance, an ANCOVA was conducted with pre-test as the covariate, which confirmed that the intervention effect remained highly significant and large, *F*(1, 275) = 402.21, *p* < .001, partial *η*^2^ = 0.594, as Table 14 illustrates.

Taken together, these analyses’ outcomes corroborate the robustness and replicability of MKIT’s instructional effects in a larger, quasi-experimental sample.

### 4.3. Evidence of Ecological Validity and Implementation Fidelity

A core strength of the MKIT framework lies in its ecological adaptability. All three studies implemented the intervention in authentic educational settings—public kindergartens and primary schools—during regular instructional hours. The structured six-week (Study 2) and ten-week (Study 3) intervention cycles were embedded seamlessly into classroom routines with minimal disruption, enhancing ecological validity and feasibility for widespread adoption.

The adapted instructional materials (Study 1)—including illustrated stories, strategic games, guided drawing, and reflective writing—were designed to align with children’s developmental capacities, and were well-received across all participating age groups, as Figure 1 illustrates. Assessments and data collection were conducted individually in quiet, distraction-free settings, minimizing external bias and ensuring the reliability of the children’s responses.

Participants were recruited from diverse socio-educational backgrounds, and inclusion criteria ensured the representation of typically developing, monolingual Romanian children, enhancing generalizability, especially in the complex stepped-wedge design, as Figure 2 shows. Additionally, teachers and parents reported informal feedback regarding increased engagement, spontaneous strategy use, and independent problem-solving behaviors, suggesting that MKIT not only fits within but actively enhances the ecological fabric of early educational environments.

### 4.4. Evidence of Transfer and Metacognitive Discourse

In addition to quantifiable outcomes, MKIT facilitated deeper internalization and spontaneous use of metacognitive language. Qualitative interviews conducted in Study 3 revealed clear differences in reflective discourse between highest- and lowest-scoring children. Highest scorers demonstrated structured, strategic narratives, while lowest scorers showed emerging awareness. Children in the intervention group increasingly used key MK phrases unprompted (e.g., “*I checked*”, “*I planned*”, “*I needed help*”), as recorded in Table 15, evidencing the transfer of MKIT principles into autonomous thinking. These verbalizations were not observed in the case of the control participant, highlighting the specificity of MKIT’s impact.

## 5. Discussion

This research program presents compelling empirical evidence supporting the Metacognitive Knowledge Intervention for Thinking (MKIT) framework as a theoretically grounded, developmentally sensitive, and psychometrically robust system for assessing and enhancing metacognitive knowledge (MK) in early and middle childhood. Across three interrelated studies involving 458 participants, the findings converge to affirm the feasibility, scalability, and cognitive impact of MKIT in real-world educational settings. This discussion synthesizes the principal contributions of the work, situates them within the broader developmental and educational psychology literature, and addresses limitations and directions for future research.

One of the most salient contributions of this research is the demonstration that children as young as 5 years old can reliably engage with structured metacognitive interventions and show measurable, durable gains in MK following targeted instruction. This challenges long-held assumptions in cognitive developmental theory which have historically situated metacognition as a late-emerging faculty, reliant on formal operational reasoning and language sophistication. While prior research has acknowledged early forms of metacognitive sensitivity (e.g., [16]; [45]), this study advances the field by offering not only a methodologically rigorous assessment tool (McKI), but also a cultural and age validated instructional framework that yields immediate and sustained developmental gains.

The equivalence in intervention responsiveness across five- and nine-year-olds in both the pilot and large-scale studies suggests that metacognitive instruction do not need to be delayed until middle to late childhood. This provides critical evidence for the developmental accessibility of structured metacognitive curricula and supports calls for the earlier integration of similar practices in early education ([13]).

Furthermore, the intervention’s substantial effect sizes (*R*^2^ = 0.481 in Study 2; *η*^2^ = 0.423 in Study 3) indicate not only statistical significance but strong practical utility, especially considering the brevity and ecological integration of the MKIT framework. That nearly half of the variance in metacognitive outcomes could be explained by a six- to ten-week intervention embedded in regular classroom instruction is a notable achievement in cognitive intervention research. Perhaps more importantly, the MKIT framework demonstrated a compensatory effect: children with the lowest baseline MK scores exhibited the largest gains. This pattern, consistent across both the pilot and validation studies, suggests that structured metacognitive instruction can serve as a developmental equalizer, eliminating early disparities in reflective thinking that may be otherwise compounded by age and schooling. These findings are theoretically consistent with threshold models of cognitive development and pragmatically significant for equity-oriented educational design. This compensatory dynamic also refines our understanding of metacognitive growth-oriented nature. The disproportionate gains among lower-performing children imply that metacognition is not fixed or innately stratified but is responsive to targeted scaffolding. This supports the hypothesis that metacognitive levels, while emergent from broader cognitive functions, can be explicitly cultivated through structured exposure, guided reflection, and recursive practice ([38]).

From a psychometric standpoint, the McKI instrument contributes significantly to the field of early cognitive assessment. The successful adaptation, cultural validation, and structural confirmation of the tool address a notable gap in the availability of developmentally appropriate metacognitive measures for young children. Its unidimensional factor structure, strong internal consistency, and high convergent and discriminant validity support its use not only in research but also in educational diagnostics and individualized instruction planning. Methodologically, the combination of exploratory and confirmatory factor analyses, stepped-wedge and quasi-experimental designs, and mixed-effects modeling offers a high standard of internal and external validity. The repeated measures framework, with individually administered assessments conducted in ecologically valid classroom settings, ensures a balance between methodological control and real-world generalizability.

Moreover, the additional qualitative analysis reveals how metacognitive interventions facilitate the transfer of metacognitive knowledge across contexts. Similarly, a critical dimension of the MKIT framework’s success lies in its ability to foster not only performance-based gains but also qualitative internalization of metacognitive strategies. Case analyses from Study 3 demonstrate that children exposed to MKIT spontaneously transferred metacognitive vocabulary and reflective strategies into novel contexts, even outside structured instructional moments. High-scoring participants consistently articulated intentional reasoning processes, while even low scorers in the experimental group exhibited clearer and more goal-oriented reflections than control peers. This observed transfer is particularly meaningful. It suggests that MKIT does not merely train children to perform well on a given task but cultivates deeper shifts in their cognitive framing and problem-solving approaches. These findings are aligned with contemporary models of metacognition as a dynamic, cross-domain competence that can be internalized through explicit instruction and recursive engagement ([12]; [10]). In this regard, MKIT can be understood as promoting a metacognitive “stance”—an epistemic orientation to one’s thinking that transcends specific tasks or domains.

Finally, it is worth mentioning that the equity implications of MKIT are especially promising. That the intervention produced equivalent gains across age, and disproportionately benefited children with lower baseline scores, suggests strong potential for reducing systemic gaps in reflective learning capacities. Because MK is a foundational predictor of academic achievement, especially in reading comprehension, problem-solving, and mathematics ([39]), early intervention in MK may yield long-term academic benefits that extend beyond the duration of the intervention itself ([43]). Importantly, the intervention is not resource-intensive. Its low-tech, teacher-deliverable structure and its compatibility with regular curricular activities enhance its scalability. The framework thus responds to dual imperatives in early education: cultivating higher-order thinking while maintaining accessibility, age-relevance and equity. Moreover, the structured integration of person, task, and strategy components—drawn from the well-established metacognitive knowledge taxonomy ([16]), and informed the selection of the instruments applied, such as the McKI and MAI—ensures that the instructional content is theoretically coherent and pedagogically comprehensive. By aligning assessment and instruction within a unified framework, MKIT allows educators to diagnose current levels of metacognitive understanding and target specific areas for development.

Despite its strengths, this research also reveals several important limitations that warrant consideration and offer valuable directions for future investigation.

First, while the samples were demographically diverse within the Romanian context, cross-cultural generalizability remains to be established. Future studies should examine the applicability and adaptability of the MKIT framework in other linguistic, cultural, and educational contexts. Given the framework’s reliance on verbal scaffolding, additional validation with linguistically diverse or multilingual populations is warranted.

Second, while the longitudinal design spanned several weeks, longer-term follow-up is needed to determine the long-term persistence of gains and their translation into academic achievement. Future work might track MKIT participants over a full academic year or link metacognitive gains to standardized academic assessments to examine downstream educational outcomes.

Third, while qualitative analysis offered valuable insights into the depth of internalization, it was limited to a small number of case studies. A more extensive qualitative inquiry—perhaps using discourse analysis or classroom observation—would enrich our understanding of how metacognitive thinking manifests spontaneously in peer interactions, play, and collaborative learning.

Consequently, while MKIT demonstrated general effectiveness, future work should explore differentiated implementation strategies that respond to individual profiles of cognitive and emotional development. This includes examining how MKIT interacts with executive function, motivation, and self-regulation, and whether these interactions vary across children with different learning needs.

## 6. Conclusions

Finally, this research provides converging evidence that the Metacognitive Knowledge Intervention for Thinking (MKIT) is a feasible and scalable framework for fostering metacognitive knowledge in children aged 5 and 9. Across three studies, evidence supported valid score interpretations, demonstrating psychometric soundness, ecological applicability, and consistent instructional effects. Notably, children with lower initial levels benefited most, indicating MKIT’s potential to reduce early achievement gaps, while comparable gains at ages 5 and 9 confirm its developmental reach. Overall, MKIT emerges as both a research instrument and an educational tool, with future work needed to extend cross-cultural validation, assess long-term impact, and tailor implementation to diverse learner profiles.

## Figures and Tables

**Figure 1 jintelligence-13-00149-f001:**
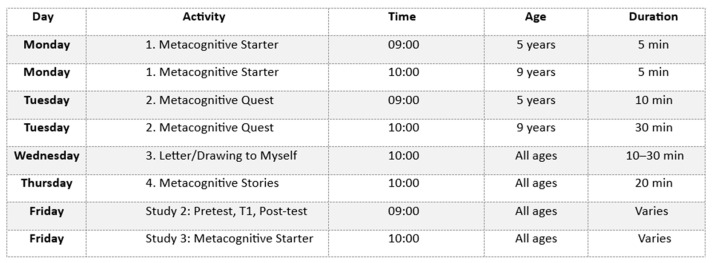
Weekly MKIT activity schedule (Study 2; Study 3).

**Figure 2 jintelligence-13-00149-f002:**
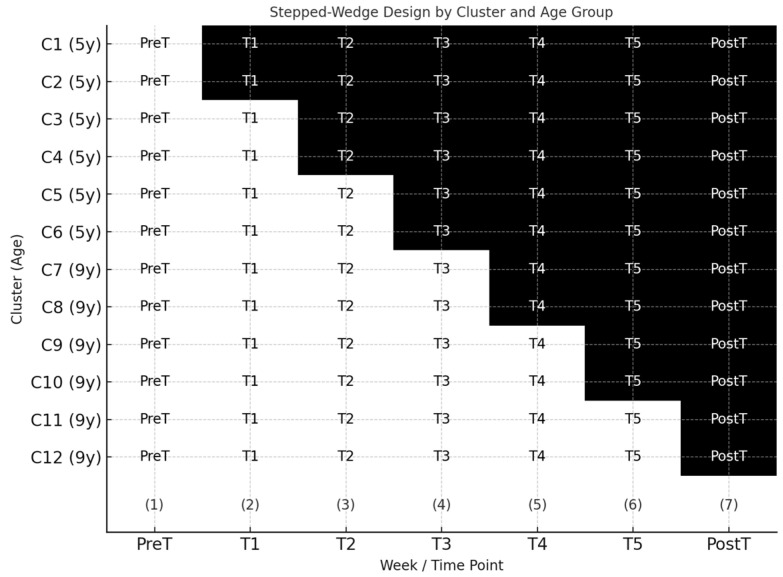
Stepped-wedge trial design with cluster crosspoint schedule and data collection timeline.

**Table 1 jintelligence-13-00149-t001:** Sampling adequacy and factor analysis suitability.

Measure	Value
Kaiser–Meyer–Olkin (KMO)	0.753
Bartlett’s Test *χ*^2^ (df = 55)	208.874
*p*-value	<0.001

**Table 2 jintelligence-13-00149-t002:** EFA and factor loadings for communalities for McKI items.

Item	Initial	Extraction
Item McKI 1	0.119	0.068
Item McKI 2	0.163	0.152
Item McKI 3	0.247	0.181
Item McKI 4	0.085	0.062
Item McKI 5	0.279	0.273
Item McKI 6	0.339	0.285
Item McKI 7	0.308	0.293
Item McKI 8	0.340	0.326
Item McKI 9	0.393	0.239
Item McKI 10	0.269	0.222
Item McKI 11	0.520	0.520

**Table 3 jintelligence-13-00149-t003:** Internal consistency indices.

Reliability Metric	Value
Cronbach’s Alpha (α)	0.760
McDonald’s Omega (ω)	0.758
Average Inter-item Correlation	0.22

**Table 4 jintelligence-13-00149-t004:** Convergent validity with MAI.

Instruments Compared	Pearson r	*p*-Value
McKI–MAI	0.738	<0.01

**Table 5 jintelligence-13-00149-t005:** Group comparison for discriminant validity.

Group	*M*	*SD*
9 years old	1.25	0.18
5 years old	0.91	0.18
Test	Value	—
*t*(98)	−9.04	—
*p*	<0.001	—
Mean difference	0.34	—
95% CI	[−0.41, −0.26]	—

**Table 6 jintelligence-13-00149-t006:** (ROC) curve for McKI total.

Reliability Metric	Value
Area under the curve (AUC)	0.962
Standard error (SE)	0.017
Significance (*p*)	<0.001
95% Confidence Interval	[0.929, 0.994]

**Table 7 jintelligence-13-00149-t007:** Confirmatory factor analysis model fit indices.

Fit Index	Value
Chi-square (*χ*^2^)	39.19
Degrees of freedom (df)	44
*p*-value	0.677
CFI (Comparative Fit Index)	1.000
TLI (Tucker–Lewis Index)	1.005
RMSEA (Root Mean Square Error of Approximation)	0.000
RMSEA 90% CI	[0.000, 0.033]
RMSEA *p*-close	0.998
*SRMR* (Standardized Root Mean Square Residual)	0.024
GFI (Goodness of Fit Index)	0.975
McDonald’s Fit Index (MFI)	1.009
Hoelter’s Critical N (α = 0.05)	430
AIC (Akaike Information Criterion)	5568.67
BIC (Bayesian Information Criterion)	5648.47
ECVI (Expected Cross-Validation Index)	0.299

**Table 8 jintelligence-13-00149-t008:** Descriptive statistics and paired samples *t*-Tests for McKI scores.

Variable	*M*	*SD*	*SE*
McKI Pre-intervention	10.19	2.96	0.125
McKI Pre-intervention	20.68	1.89	0.080

**Table 9 jintelligence-13-00149-t009:** Paired samples *t*-Tests results.

Pair	Mean Difference	*SD*	*SE*	*t*	df	*p*
McKI Pre-intervention	−10.487	3.161	0.134	−78.510	559	0.001
McKI Pre-intervention	—	—	—	—	—	—

**Table 10 jintelligence-13-00149-t010:** Tests of between-subjects effects for age and intervention and their interaction (Study 2).

Source	Type III Sum of Squares	df	Mean Square	*F*	Sig.
Corrected Model	50.464	3	16.821	171.809	0.000
Intercept	846.158	1	846.158	8642.555	0.000
Age	0.001	1	0.001	0.011	0.917
Intervention Status	50.266	1	50.266	513.406	0.000
Age × Intervention	0.159	1	0.159	1.620	0.204
Error	54.436	556	0.098	—	—
Total	1154.917	560	—	—	—
Corrected Total	104.899	559	—	—	—

**Table 11 jintelligence-13-00149-t011:** Means and standard deviations for gain scores by baseline performance group (Study 2).

Baseline Performance Group	n	M (Gain)	*SD*
Low	56	−14.50	3.07
Medium	266	−11.92	2.11
High	161	−8.61	1.64
Very high	77	−6.55	1.79

**Table 12 jintelligence-13-00149-t012:** Bonferroni post hoc test results for MKIT gain scores (Study 2).

Initial Group (I)	Compared Group (J)	M Diff (I − J)	SE	*p*	IC 95% Lower	IC 95% Upper
Low	Medium	−2.579	0.34	<0.001	−3.38	−1.77
	High	−5.891	0.30	<0.001	−6.74	−5.04
	Very high	−7.955	0.33	<0.001	−8.92	−6.99
Medium	Low	2.579	0.34	<0.001	1.77	3.38
	High	−3.312	0.26	<0.001	−3.86	−2.77
	Very high	−5.376	0.27	<0.001	−6.08	−4.67
High	Low	5.891	0.30	<0.001	5.04	6.74
	Medium	3.312	0.206	<0.001	2.77	3.86
	Very high	−2.063	0.286	<0.001	−2.82	−1.31
Very high	Low	7.955	0.363	<0.001	6.99	8.92
	Medium	5.376	0.267	<0.001	4.67	6.08
	High	2.063	0.286	<0.001	1.31	2.82

**Table 13 jintelligence-13-00149-t013:** Descriptive statistics and paired samples t-Tests for McKI scores (Study 3).

Group	Time	Mean	Standard Deviation	N
Intervention	Pre-test	11.252	6.262	218
Intervention	Post-test	14.537	5.402	218
Control	Pre-test	7.467	1.836	60
Control	Post-test	7.467	1.836	60

**Table 14 jintelligence-13-00149-t014:** ANCOVA results for MKIT intervention effect on McKI scores (Study 3).

Source	SS	df	MS	*F*	*p*	Partial *η*^2^
PreTotal	6075.13	1	6075.13	3663.64	<.001	0.930
Group	666.96	1	666.96	402.21	<.001	0.594
Error	456.01	275	1.66	—	—	—

**Table 15 jintelligence-13-00149-t015:** MK features and transfer sample phrases by group.

Participant Group	MK Narrative Features	Example Spontaneous Phrases
Highest Score Experimental	Structured, strategic, goal-oriented; used clear planning and review	“I planned my steps”; “I checked my work”; “I verified”; “I compared the result”; “I said to myself”; “I thought to myself/that/if”;
Lowest Score Experimental	Emerging awareness; partial references to task steps; less coherent	“I needed help”; “I just know how to do it”; “I know when is not correct”;
Medium Score Control	Lacked reflective language; task-focused responses only	“I just did it” (no metacognitive language)

## Data Availability

The data presented in this study are available on request from the corresponding author. The data are not publicly available as they are part of an ongoing doctoral research project and are reserved for further dissertation purposes.

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
