# Peer review of "The Meta-Intelligent Child: Validating the MKIT as a Tool to Develop Metacognitive Knowledge in Early Childhood"

_jintelligence, 2025, doi:10.3390/jintelligence13110149_

Round 1
Reviewer 1 Report
Comments and Suggestions for Authors
The article entitled “The Meta-intelligent Child: Validating the MKIT as a Tool to Develop Metacognitive Knowledge in Early Childhood” validates a tool for assessing metacognition in young Romanian children.
The article is well written, with the various stages clearly detailed, which makes it possible to replicate the results for those interested. However, here are a few comments to help improve the manuscript:
The literature review would benefit from being expanded (only 34 references) and made more relevant by drawing on more recent articles (no references from the last 5 years, 23 from the last 20 years, and 11 from even earlier).
It would be relevant to explain the reasons that led the authors to conduct their studies only with children aged 5 and 9 (and why with a developmental aspect for children aged 5 to 9).
In the empirical parts of the studies (I would like to emphasize once again the detailed procedure provided), the authors chose to present the results not only within each study but also in an “integrated results” section. This seems redundant, as the results of the three studies are repeated.
Author Response
Honored reviewer,
We would like to thank you and the reviewers for the thoughtful and constructive feedback on our manuscript. We carefully considered each comment and have revised the manuscript accordingly. Below we provide a detailed point-by-point response, with an explanation of the changes made. Revisions in the manuscript are highlighted in the tracked version: changes addressing Reviewer 1 are marked in blue.
Comment 1: “The literature review would benefit from being expanded (only 34 references) and made more relevant by drawing on more recent articles (no references from the last 5 years, 23 from the last 20 years, and 11 from even earlier).”
Response:
We thank the reviewer for all helpful observations. We expanded the literature review by incorporating several recent studies published within the last 5 years, particularly in the areas of metacognition, early childhood assessment, and validation methodologies. The revised section now includes both seminal works and up-to-date references to strengthen the theoretical and empirical foundation.
Changes made: 1. (Appelbaum, 2023); 2. (Buehler, 2025); 3. (Chen, 2022); 4. (Eberhart, 2024); 5. (Frumos, 2024); 6. (Goretzko, 2025); 7. (Kolloff & colleagues, 2024); 8. (Lorenzo-Seva, 2024); 9. (Marulis, 2025); 10. (McNeish & Wolf, 2023); 11. (Ossa, 2024); 12. (Shields, 2024); 13. (Susnea, 2025).
Comment 2: “It would be relevant to explain the reasons that led the authors to conduct their studies only with children aged 5 and 9 (and why with a developmental aspect for children aged 5 to 9).”
Response:
We indeed, have clarified our rationale for selecting children aged 5 to 9.
Changes made: Added explanation in Introduction Section (rows 101-113 and rows 159-164) and changed within the text from 5 to 9 to 5 and 9.
Comment 3: “In the empirical parts of the studies (I would like to emphasize once again the detailed procedure provided), the authors chose to present the results not only within each study but also in an ‘integrated results’ section. This seems redundant, as the results of the three studies are repeated.”
Response: To address this and avoid redundancy, we streamlined the presentation.
At the same time, we also carefully must consider another reviewer’s observation regarding the need to frame the results within the logic of validation research (Messick, 1995; Kane, 2013), which emphasizes the integration of different strands of evidence into a unified validity argument. In light of both suggestions, we have revised the structure: the previous “Integrated Results” section has been reframed as part of the “Discussions”. In this revised version, we focus on synthesizing the overarching findings without repeating details already presented, while still making explicit how the results contribute to the process of validation.
Changes made: Section 4 (formerly Integrated Results) has been retitled and reframed as Discussions, emphasizing synthesis and alignment with validity theory, but also removing redundancies, as required.

Reviewer 2 Report
Comments and Suggestions for Authors
This research validated the Metacognitive Knowledge Intervention for Thinking (MKIT) to assess and develop domain-general metacognitive knowledge (MK) in children aged 5 to 9. Study 1 focused on the cultural adaptation and preliminary psychometric validation of the McKI for use with Romanian children; Study 2 piloted the full MKIT intervention using a stepped-wedge experimental design; Study 3 served as the confirmatory phase of the research program and aimed to test the generalizability and full-scale effectiveness of the MKIT framework across a larger sample of children. The research results indicated MKIT is a theoretically grounded tool for assessing and enhancing metacognitive knowledge (MK) in early and middle childhood.
This research was well-designed. In each study, the hypotheses were verified one by one, and the statistics were appropriately used to answer the research questions. The authors demonstrated a professional academic background and strong research capabilities. Basically, this is an outstanding paper. There are two suggestions:
- The ethics review should state the name of the review institution and the reference number. In addition, are the three studies under the same review case?
- In study 2, Baseline-Dependent Effects mentioned that children with lower baseline metacognitive scores experienced greater relative benefits from the intervention (F=250.97, p<.001). In which table can the readers find this data?
- In Table 11, why was there such a large gap between the pretest scores of the intervention group and the control group? Should the experiment be conducted with two groups of children whose pretest abilities were more similar?
- Are there two Table 11? Is the table in 4.4 Table 12?
-
Author Response
Honored Reviewer,
We would like to thank you and the reviewers for the thoughtful and constructive feedback on our manuscript. We carefully considered each comment and have revised the manuscript accordingly. Below we provide a detailed point-by-point response, with an explanation of the changes made. Revisions in the manuscript are highlighted in the tracked version: changes addressing Reviewer 1 are marked in green.
Comment 1: “The ethics review should state the name of the review institution and the reference number. In addition, are the three studies under the same review case?”
Response: We sincerely thank the reviewer for pointing this out. We have now included the name of the ethics review institution and the reference number directly in the manuscript to ensure full transparency. All three studies were conducted under the same ethical review case. As indicated to the Editor-in-Chief, the official ethics approval document with signature and reference number has also been provided.
Changes made: Provided the Editor-in-Chief, with institution name and ethics approval reference number as well as with the official document.
Comment 2: “In study 2, Baseline-Dependent Effects mentioned that children with lower baseline metacognitive scores experienced greater relative benefits from the intervention (F = 250.97, p < .001). In which table can the readers find this data?”
Response: We are grateful for the reviewer’s careful reading and for identifying this omission. We have now corrected this oversight by explicitly linking the reported finding to the appropriate tables. The data are now presented in the revised manuscript in Table 11 and Table 12.
Changes made: Section 4. (Discussions) now includes direct references to Tables 11 and 12.
Comment 3: “In Table 11, why was there such a large gap between the pretest scores of the intervention group and the control group? Should the experiment be conducted with two groups of children whose pretest abilities were more similar?”
Response: We thank the reviewer for raising this important point. The observed gap is due to differences in group sizes between the intervention and control groups. To clarify this, we added explanatory notes in the manuscript and included additional ANCOVA analyses that adjusted the baseline differences. These supplementary analyses confirm that the intervention effects remain robust.
Changes made: Clarifications added in Section 4. (Discussions), with supporting analyses reported in the newly added Tables 13 and 14.
Comment 4: “Are there two Table 11? Is the table in 4.4 Table 12?”
Response: The duplicated numbering has been resolved, and all tables are now correctly labeled.
Changes made: All tables renumbered consistently across the manuscript.

Reviewer 3 Report
Comments and Suggestions for Authors
Thank you for the opportunity to review this paper. It was generally well written especially the literature review and set up.
The following suggestion are meant to inform authors on how to improve the paper and focus more on methodological issues.
- I am unsure or rather unclear about the using 50 5 year olds and 50 9 year olds - earlier in the paper it said this toolkit is for 5-9 year olds. So using the extremes in the first study was a way to bracket this - was this intentional- how does this age range relate theoretically. Why use extremes and not people aged 6, 7 and 8 to see progressions?
- I believe the authors should focus on the broader conception of validity as a whole and not these different parts of validity evidence as buckets and I believe the authors should provide a deep rationale for each of these tests using measurement literature.
- I did feel the paper was very devoid of current methodological citations to help justify decisions.
- The use of hypothesis for a validity study was different and this framing felt off as validation studies generally do not frame their papers in this manner- I recommend reading and using validation studies as mentor texts for the framing.
- The Present Study section offered very theoretical and broad epistemological contributions to the literautre- I felt confused and that this section over promised what was delivered and there was not adequate followup to provide how the paper did this- I would recommend removing some of that.
- Using the MAI to get evidence of validity needs to be justified more thoroughly and with the appended tables and figures I was not convinced that these items aligned the way the table stated- it felt a bit random.
Author Response
Honored Reviwer,
We would like to thank you and the reviewers for the thoughtful and constructive feedback on our manuscript. We carefully considered each comment and have revised the manuscript accordingly. Below we provide a detailed point-by-point response, with an explanation of the changes made. Revisions in the manuscript are highlighted in the tracked version: changes addressing Reviewer 1 are marked in yellow.
Comment 1: “I am unsure or rather unclear about the using 50 5 year olds and 50 9 year olds – earlier in the paper it said this toolkit is for 5–9 year olds. So using the extremes in the first study was a way to bracket this – was this intentional? How does this age range relate theoretically? Why use extremes and not people aged 6, 7 and 8 to see progressions?”
Response: We are grateful for all important observations. We clarified that using ages 5 and 9 was intentional to bracket the developmental range targeted by the toolkit. Age 5 (pre operaÈ›ional) represents the onset of metacognitive awareness, while age 9 ( concrete operational) represents its consolidation in early schooling. This design maximizes developmental contrast for initial validation. We also acknowledge that including ages 6–8 would allow for testing progressions, which we identify as a direction for future research.
Change made: Clarification added in Section 3.2.1 (rows 465-468).
Comment 2: “I believe the authors should focus on the broader conception of validity as a whole and not these different parts of validity evidence as buckets and I believe the authors should provide a deep rationale for each of these tests using measurement literature”.
Response: We sincerely thank the editor for this valuable comment. We have revised the framing to align explicitly with Messick’s (1995) unified theory of validity and Kane’s (2013) argument-based validation, making clear that the different strands of evidence (psychometric analyses, intervention effects, qualitative data) all contribute to a single validity argument. At the same time, in order to make the research more accessible to readers interested in replication of the intervention without necessarily engaging in the validation framework, and in response to other reviewers’ requests, we are contioned to retained the use of “hypotheses” as a structuring mechanism. As can be observed in the revised manuscript, the results are presented succinctly within each study, while the integrated set of tables and the final conclusive synthesis are gathered just at the end, in Section 4 (Discussion).
Change made: Text revised in Section 2 (The present research, where we explained the validation framework explicit ) and Section 4 (Discussion).
Comment 3: “I did feel the paper was very devoid of current methodological citations to help justify decisions”.
Response: Thank you! We fully agree and have expanded the methodological grounding. We added both classic and recent methodological references for EFA (Horn, 1965; Lorenzo-Seva, 2024), CFA (Hu & Bentler, 1999; Goretzko et al., 2025), reliability (Sijtsma, 2009; McNeish & Wolf, 2023), and measurement invariance (Meredith, 1993; Putnick & Bornstein, 2016).
Change made: References inserted in text (color yellow), as well as in Appendix.
Comment 4: “The use of hypothesis for a validity study was different and this framing felt off as validation studies generally do not frame their papers in this manner – I recommend reading and using validation studies as mentor texts for the framing”.
Response: We appreciate this observation. We clarified now that what we termed “hypotheses” were not intended as causal predictions but as validation expectations guiding the analytic strategy. This framing is now explicitly linked to Kane’s (2013) argument-based approach and the Standards for Educational and Psychological Testing.
Change made: Clarifications added in Section 3.1.2 (Design, procedure and psychometric expectations).
Comment 5: “The Present Study section offered very theoretical and broad epistemological contributions to the literature – I felt confused and that this section overpromised what was delivered and there was not adequate follow-up to provide how the paper did this – I would recommend removing some of that”.
Response: We thank the editor for this guidance. We streamlined The Present Study section to make it concise and concrete. Broader epistemological claims were removed.
Change made: Section The Present Study rewritten, words as “epistemological” removed or replaced.
Comment 6: “Using the MAI to get evidence of validity needs to be justified more thoroughly and with the appended tables and figures I was not convinced that these items aligned the way the table stated – it felt a bit random”.
Response: Thank you for this important point. We revised Section 3.1.1. and Appendix A.1, to include the new additions and clarifications to the alignment.
More important, must added here that we chosed MAI, to avoid testing convergent validity by comparing two instruments that are not validated in Romanian. We selected the MAI, which is already translated and validated. Currently, very few instruments—especially for children—are available in Romanian with full statistical validation, making the MAI the most methodologically sound choice. This decision is also supported by prior research: as suggested by Schraw & Dennison (1994), the MAI is widely recognized for its ability to capture both knowledge about cognition and regulation of cognition. More recent work (e.g., Sperling et al., 2002) also highlights the MAI’s flexibility in being adapted across different age groups and educational contexts. Yet, the most important reference here is Henter (2016), which presents a structure where these items are explicitly grouped and proven to measure knowledge about cognition for romanian population. These are precisely the items we have integrated and presented fully in the revised version.
Change made: Section 3.1.3. (rows 374-377) and Appendix A.1 was updated accordingly.
We sincerely thank you for your insightful and constructive feedback. We believe the revision have substantially improved the clarity and contribution of our manuscript, so thank you!
Sincerely

Round 2
Reviewer 1 Report
Comments and Suggestions for Authors
I would like to thank the authors for the changes made to the first parts of the document. The literature review is a little more detailed (although there is one caveat, as the authors straddle the gap between very recent references and very old references, which gives a strange impression).
I have no specific comments to add regarding the procedure.
Regarding section 4, initially titled “integrated results,” the authors chose to rename it “discussion” even though they still present results in it. Then, they renamed section 5 ‘conclusion’ to “discussion.” And again, this does not correspond to what is presented, which is a detailed discussion of the results in relation to recent scientific literature, followed by a presentation of the implications for research and for professionals in the field, before concluding.
Author Response
Comment 1:
I would like to thank the authors for the changes made to the first parts of the document. The literature review is a little more detailed (although there is one caveat, as the authors straddle the gap between very recent references and very old references, which gives a strange impression).
Response:
We sincerely thank the Reviewer for acknowledging the improvements in the literature review. We also carefully revised the references to reduce the impression of imbalance between very recent and very old sources. We retained a few classical works that remain theoretically foundational (e.g., Flavell, 1979), but we have now complemented them with more mid-range and recent references to ensure better chronological and conceptual balance. These updates are highlighted in green in the revised manuscript.
Comment 2:
I have no specific comments to add regarding the procedure.
Response:
We thank the Reviewer for this acknowledgment.
Comment 3:
Regarding section 4, initially titled “integrated results,” the authors chose to rename it “discussion” even though they still present results in it. Then, they renamed section 5 ‘conclusion’ to “discussion.” And again, this does not correspond to what is presented, which is a detailed discussion of the results in relation to recent scientific literature, followed by a presentation of the implications for research and for professionals in the field, before concluding.
Response:
We are grateful for this observation, which guided us to clarify and restructure the manuscript. In addition, we were specifically asked to adopt Messick’s unified perspective on instrument validity, which treats validity not as separate types but as a single construct supported by multiple strands of evidence. This conceptual shift also required us to ensure that findings from each study (Studies 1–3) are presented progressively, but also integrated into a coherent, interconnected argument about validity.
Accordingly, in the revised version:
- Section 4 is now titled Integrated Findings of Validity and brings together the different strands of evidence across 3 different studies (structural, consequential, generalization, transfer), presenting them in a unified way consistent with Messick’s framework.
- Section 5 is now titled Discussion and is fully dedicated to interpreting results, situating them in the broader literature, and discussing implications and limitations.
- Section 6 – a new small section was introduced, titled Conclusions, providing a concise summary.
These revisions ensure that the structure of the article is now fully consistent with both the Reviewer’s recommendation and the conceptualization of validity as an ongoing process supported by converging evidence. All these modifications resolve the mismatch between section titles and content, and they are highlighted in green in the revised manuscript.
Reviewer 2 Report
Comments and Suggestions for Authors
All suggestions have been modified and provided with relevant information, and they have been approved for publication.
Author Response
Comment: All suggestions have been modified and provided with relevant information, and they have been approved for publication.
Response to Reviewer
We sincerely thank the Reviewer for all the feedback. We greatly appreciate the time and effort dedicated to reviewing our work. Sincerely
Reviewer 3 Report
Comments and Suggestions for Authors
I really appreciated the inclusion of Messick and Kane when discussing validity theory. Thank you for this. However, then in the results, results were still discussed as separate types of validity, negating what was mentioned earlier in how we conceptualize this concept of validity.
You should also go through the paper thoroughly and anywhere we you claim the toolkit is valid- rewrite to state shows evidence of validity in making X inferences. Validity is not a static property.
The paper and the presentation of findings needs to be thoroughly combed over to ensure that the presentation of the results fits in with the current conceptualization of validity.
Author Response
Comment:
I really appreciated the inclusion of Messick and Kane when discussing validity theory. However, then in the results, results were still discussed as separate types of validity, negating what was mentioned earlier in how we conceptualize this concept of validity. You should also go through the paper thoroughly and anywhere you claim the toolkit is valid—rewrite to state shows evidence of validity in making X inferences. Validity is not a static property. The paper and the presentation of findings needs to be thoroughly combed over to ensure that the presentation of the results fits in with the current conceptualization of validity.
Response:
We thank the Reviewer for this insightful guidance. In the revised manuscript, we aligned the entire validation argument with Messick’s unified framework and Kane’s argument-based approach. Specifically:
- No longer presenting “types” of validity.
Throughout the results and synthesis, we replaced references to separate “types” with strands of evidence supporting valid score interpretations (e.g., content/response-process, internal structure, relations to other variables, generalization, consequences). This removes any implication that validity is partitioned into static categories and reflects a progressive and unified argument. - Rewriting claims about the toolkit.
We systematically replaced formulations such as “the tool/instrument is valid” with statements like “the evidence supports valid inferences from MKIT/McKI scores for the intended uses”. Where appropriate, we explicitly state what inferences are warranted, for example: - distinguishing age-related differences in MK,
- monitoring change over time following instruction,
- informing classroom practice by identifying areas of MK that can be strengthened.
This reframing emphasizes that validity concerns the interpretation and use of scores, not a static property of the instrument. - Ensuring consistency with the contemporary view of validity.
We conducted a line-by-line revision of the Abstract, the Validation Framework subsection, each study’s results, the integrated synthesis, and the Conclusions to maintain consistent argument-based phrasing (e.g., “evidence supporting valid score interpretations” rather than claims of validity-as-attribute). - Integrating evidence across studies to reflect validity as an ongoing process.
To avoid treating findings in isolation, we renamed Section 4: Integrated Findings of Validity, which weaves together the isolated strands of evidence (of each Studies 1–3) into a single, interconnected validity consistent argument. - Concrete wording updates (examples).
- Before: “The instrument is valid and reliable for early metacognitive assessment.”
After: “The accumulated evidence supports valid interpretations of McKI scores for early metacognitive assessment, with acceptable internal consistency and a coherent latent structure.”
All revisions addressing this comment are highlighted in blue in the manuscript. We appreciate the Reviewer’s direction; it substantially improved the theoretical coherence and reporting consistency of our validity argument.
Round 3
Reviewer 1 Report
Comments and Suggestions for Authors
I thank the authors for the changes made
Reviewer 3 Report
Comments and Suggestions for Authors
Thank for you addressing the revisions.